# Experimental and Numerical Study on the Influence of Lubrication Conditions on AA6068 Aluminum Alloy Cold Deformation Behavior

**DOI:** 10.3390/ma16052045

**Published:** 2023-03-01

**Authors:** Mariana Florica Pop, Adriana Voica Neag, Ioana-Monica Sas-Boca

**Affiliations:** Faculty of Materials and Environmental Engineering, Technical University of Cluj Napoca, 28 Memorandumului Street, 400114 Cluj-Napoca, Romania

**Keywords:** cold deformation, friction coefficient, metal forming, simulation (FEM)

## Abstract

The aim of this manuscript is the experimental and numerical study regarding the influence of friction conditions on plastic deformation behavior by upsetting the A6082 aluminum alloy. The upsetting operation is characteristic of a significant number of metal forming processes: close die forging, open die forging, extrusion, and rolling. The purpose of the experimental tests was to determine: by the ring compression method, the friction coefficient for 3 surface lubrication conditions (dry, mineral oil, graphite in oil) by using the Coulomb friction model; the influence of strains on the friction coefficient; the influence of friction conditions on the formability of the A6082 aluminum alloy upsetted on hammer; study of non-uniformity of strains in upsetting by measuring hardness; change of the tool-sample contact surface and non-uniformity of strains distribution in a material by numerical simulation. Regarding the tribological studies involving numerical simulations on the deformation of metals, they mainly focused on the development of friction models that characterize the friction at the tool-sample interface. The software used for the numerical analysis was Forge@ from Transvalor.

## 1. Introduction

The plastic deformation technology of metals occupies a special place in processing technologies. Among the advantages of processing by plastic deformation, it can be mentioned the obtaining of superior mechanical properties, with minimal waste of material. In the case of plastic deformation processes, the starting material has a simple geometry. The material is plastically deformed in several stages, obtaining a product with a complex configuration. Achieving net-shape or near-net-shape products reduces metal removal requirements, resulting in significant material and energy savings. The pressure transmitted from the dies to the workpiece determines the flow of the metal. The increase in ductility and decrease in yield strength of the material occurs when the deformation temperature increases [1].

In metal forming processes, will occur friction, because between two or more surfaces will always be contact. Friction includes parameters that interact such as lubrication condition, surface roughness, sliding speed, temperature, contact pressure, material properties.

The material flow, surface quality and tool life are influenced by the friction between tool and workpiece [2].

Aluminum alloys are being used for applications in aerospace and automobile industries, main benefits of this alloy being: low density, good resistance to corrosion, good conductor of heat and electricity; easily machined and deformed using a wide variety of deformation processes, not least being easy to recycle. Aluminum 6068 is widely used for manufacturing aircraft structures, fuselages, wings, automobile parts and other types of components [3].

Male A.T. and Cockcroft M.G. [4] were the first researchers to study the coefficient of friction in plastic deformation. The research was continued by Mahrenholtz O. and others [5]. Sahin M. and others [6] studied the influence of surface roughness on friction. Altan T et al. [1], Kobayashi S. et al. [7], Li K. et al. [8], Srivastava A. et al. [9], Kanca, E, et al. [10], Davidson, Dr M J.and others [11], Cora, Ö. N. [12] and Kunogi M. and others [13] carried out research in the field of finite element analysis of the phenomenon of friction in plastic deformation.

Valberg S.H [14], Joun M.S., and others [15] studied the effects of friction laws on plastic deformation processes. Camacho A.M. [16] and others studied friction factors in metal forming and Trzepiecinski T. and others [17] studied developments and trends in friction testing for conventional sheet metal forming and incremental sheet forming.

Research on barrelling of aluminum solid cylinders during cold upset forging with constraints at both ends was carried out by Malayappan S et al. [18,19,20], Manisekar, K. and Narayanasamy, R. [21] and Priyadarshini A. et al. [3]. A review of friction test techniques to be studied by Wang L. et al. [22]. The effect of lubrication [23], the determination of the coefficient of friction [24,25], and the friction factor [26] have been analyzed by researchers. Research on the influence of friction on the formability of metallic materials has also been carried out by, Pop M. et al. [27,28,29].

The use of computer-aided design and manufacturing techniques has an essential role in modern material deformation technologies. Process modeling to investigate and understand deformation mechanics has become a major research concern, and the application of the finite element method (FEM) has gained a special role, especially in the modeling of forming processes. The main objectives of numerical simulation in the manufacturing process are to reduce manufacturing costs and increase product quality.

The friction coefficient in the case of the hot forging of a Ti-6Al-4V alloy was determined by combining the ring method with finite element simulation. The conclusion drawn was that the heat transfer coefficient has a significant effect on the flow of the material and therefore on the friction coefficient [30]. The application of the finite element analysis method for the study of friction in plastic deformation processes can be found in numerous studies [31,32,33].

## 2. Friction in Metal Forming

During metal deformation processes, friction is influenced by surface roughness. As can be seen in Figure 1 metal surfaces are not smooth; there are asperities of different types and different distributions, representing surface roughness [5,6]. Until all asperities are flattened, surface roughness has an influence on the frictional properties of these surfaces, especially at the beginning of metal deformation processes [6].

In metal-forming processes, the high contact pressures increase during the process, and temperatures are high, especially in the case of warm and hot forming. For these reasons, the friction coefficients have high values even when using lubricants. Friction produces undesirable phenomena such as high energy, strong die wear, increased deformation force, tool temperature increase, increased product cost, decreased product quality, etc. [15,16].

Figure 2 shows the main parameters that influence the phenomenon of friction in the metal-forming processes.

The open die forging process and especially the upsetting operation is widely used for the plastic deformation of metallic materials. Also, compression testing along with traction and twisting is part of the techniques used to determine the deformation behavior of materials. The compression test allows, among others, the determination of the yield stress, and the strain to fracture. The friction conditions influence the deformation behavior and the flow of materials during the deformation processes, especially in the case of compression.

A workpiece with a cylindrical, square, rectangular cross-section subjected to compression between two flat surfaces will undergo a heterogeneous deformation. The material that comes into contact with the two flat surfaces will suffer the smallest deformation, being restricted in movement by the presence of frictional forces. The material in the central part of the specimen which does not suffer from any frictional resistance will flow freely in the outward direction. Such a phenomenon is called barreling. The degree of bulging can be reduced by the correct use of lubricants [18,19,20,21,22,23,24,25,26].

The appearance of frictional forces during the compression deformation of a cylindrical semi-finished product leads to the formation of 3 areas, as can be seen in Figure 3.

The most used method for determining the coefficient of friction is the ring compression test. The idea of the test is to increase or decrease the inner diameter of a ring sample, when it is compressed between plane plates. It provides special knowledge about the coefficient of friction at the part-die interface. If friction is low (good lubrication), the inner diameter increases; while if the friction is high (poor lubrication), the internal diameter decreases as shown in Figure 4 [9,15,16,22,23,24,25,26].

This method of determining the coefficient of friction was first used by Kunogi [13] and then developed by Male and Cockcroft [4].

Numerous theoretical models have been developed to study the phenomenon of friction in plastic deformation processes. Among these, the most used are:−Coulomb’s model is expressed as follows:
(1)τ=μp
where *τ* is the shear friction coefficient, *μ* is the friction coefficient between the surface of the workpiece and the die, and *p* is the normal force applied on the surface. This model is valid in the case of deformation processes in which the deformation pressures are relatively low [4,7].−Shear Friction Model expresses is as follows:
(2)τ=mk
where *m* is the shear friction factor, and *k* is the shear yield limit.

The model is specific to deformation processes with high pressures.

Figure 5 presents a typical friction calibration curve, which graphically shows the variation of the inner diameter reduction of the ring depending on the height reduction, which can be used to determine the friction factor *m*.

The diagram above makes the connection between the percentage reduction of the internal diameter of the sample and the percentage reduction of its height for different coefficients of friction.

## 3. Experimental Details

The material used for the present manuscript was AA6068-T6 aluminum alloy. Its chemical composition is presented in Table 1.

### 3.1. Influence of Strains on Friction Coefficient

The friction coefficients were determined by ring compression tests in the following friction conditions: dry, lubricated with mineral oil, and graphite in oil. The samples were subjected to three different strains. The equipment used for this method was a hydraulic press, with a maximum force of 200 kN. The ring dimensions are shown in Figure 6. The experiments were conducted at room temperature. All samples have approximately the same surface finish and before compression, the lubricants were deposited on the surfaces of the ring and the dies.

The results of the experiments are presented in Figure 7.

Considering the irregularities of the inner and outer surfaces of the specimen, several measurements were made and the average values of the inner and outer diameters of the ring were recorded in Table 2. In the experiments presented below, it was considered the Coulomb friction model, so the friction coefficient. Using the friction coefficient calibration curve of Cockcroft and Male [4] from Figure 5, the coefficient of friction was determined for each sample. The results are presented in Table 2.

Figure 8 shows the variation of the coefficient of friction according to strains for different friction conditions.

The graph (Figure 8) shows the reduction in the value of the friction coefficients, at degrees of deformation greater than 0.3. A more significant reduction is observed in the case of non-lubricated surfaces. This may be due to the presence of oxides that can act as a lubricant. For unlubricated surfaces, the coefficients of friction are higher, compared to surfaces lubricated with mineral oil and graphite in oil. In the case of dry surfaces, the friction coefficient values vary between 0.15 at a strain of 0.2 and 0.20 at a strain of 0.3. The minimum values of the friction coefficient in the case of graphite in oil lubrication vary between 0.055 at a strain of 0.2 and 0.09 at a strain of 0.4.

It can also be observed that at strains greater than 0.3, there is a tendency to stabilize the value of the friction coefficient, especially in the case of mineral oil and graphite in oil lubrication.

### 3.2. Influence of Friction Conditions on Material Formability

To study the influence of friction conditions on the material formability, three series of samples were deformed on hammering, with different ram drop heights, so that the deformation energies were different. Ram drop heights were: 1850, 1500, and 1000 mm.

Cylindrical specimens of 18 mm diameter and 30 mm height, Figure 9, were prepared from AA6068-T6 bar using machining operations on a lathe.

Figure 10a–i shows the shape of the samples after deformation.

To determine the material’s strength to deformation the following formulas were used:(3)$$\sigma_{d}=\frac{L}{mVln\frac{h_{0}}{h}}\;\;\left[N/{\mathrm{mm}}^{2}\right]$$
where:h_0_, d_0_ are the initial dimensions of the sample [mm];*H_s_* is the fall height of the ram [mm];*L* is the mechanical work of deformation [daN·mm];*h* is the height of the sample after deformation [mm];*d_min_*, *d_max_* are the minimum and maximum diameters of the deformed sample.m—factor.
(4)$$m=1+0.1\frac{d_{min}}{h}$$

Three types of lubricants were used for each test series to ensure different friction conditions.

The condition of the surfaces was as follows: dry, mineral oil lubricated and graphite in oil lubricated surfaces. After deformation, the height, minimum and maximum diameter for each sample were measured and reported in Table 3.

The friction coefficient was determined with the relation of S. I. Gubkin [13]:(5)$$\mu =\frac{6.25\left(\delta +2\delta^{2}\right)}{1+\varepsilon }{\left(\frac{d_{0}}{h_{0}}\right)}^{1.5}$$
where *δ* is barreling coefficient calculated with the formula:(6)$$\delta =\frac{d_{max}-d_{min}}{d_{max}}$$

The obtained results are presented in Table 4.

Based on the experimental results, the graphs in Figure 11 and Figure 12 were drawn, representing the variation of the deformation strength vs. deformation energy, respectively the variation of the friction coefficient function on the strain.

The material strength decreases with increasing energy for all friction conditions. For dry surfaces, the strength is higher and for lubricated surfaces with mineral oil and graphite, it is lower, which means that lubrication of the surfaces has a significant influence on the material strength.

In Figure 12, the variation of the friction coefficient function on strain is highlighted. The friction coefficient increases with increasing strain for all friction conditions.

For the unlubricated samples the highest friction coefficients were obtained, and for surfaces lubricated with mineral oil and graphite, the friction coefficients are much lower.

## 4. Strains Non-Uniformity in the Samples Using Hardness Measurement Method

This method is only applicable to cold plastic deformation and consists of deforming the sample, slicing, and measuring the hardness in 3 different areas, Figure 13.

In the image below Figure 14, cylindrical samples deformed in different conditions of deformation and lubrication are shown, sliced, and prepared for hardness determination.

In the following graphs, Figure 15, Figure 16 and Figure 17, shows that the hardness varies depending on the height of the ram falling in different lubrication conditions. Graphs were plotted for each deformation area.

Area 1 has the lowest hardness values since it is the zone with the least deformation (Figure 15). As can be seen from the Figure, the hardness of the unlubricated samples is lower than that of the oil-lubricated samples and the graphite-in-oil samples because the presence of lubricant reduces friction, and the samples deform more.

In the deformation area 2, the deformation is medium, and the hardness is higher than in area 1, Figure 16. Similarly, the presence of lubricants on the surface increases hardness and deformation.

Area 3 is the most deformed area, with the highest values for hardness. In this case, the hardness reaches up to 55 HRB on surfaces lubricated with graphite in oil, Figure 17.

## 5. Numerical Analysis Formulation

The finite element method has been used to study and analyze the material behavior during the 3D upsetting process between the two flat and parallel tools. The influence of the friction phenomena on the modification of the contact area between the deformation elements and the sample during the upsetting process was studied by using TRANSVALOR FORGE NXT 3.2 software for the simulation. The entire assembly (upper die, lower die, and sample) is 3D modeled in SOLID WORKS and saved as an STL file to be used in the FEM simulation analysis. The dies are set as rigid bodies at 20 °C, and all deformations of the tools were neglected. The A6082 alloy sample had an initial height of 30 mm and a diameter of 18 mm.

The numerical simulations were performed under axial symmetry conditions and set up properly. To save calculation time and data storage space, only 1/4 of the model was investigated, considering the geometrical symmetry. A four-node tetrahedron element was applied to the sample, consisting of about 18,797 elements and 4132 nodes. The upper and lower dies were only meshed on the surface. The incremental control step is set constantly at 0.5 mm. The software uses an incrementally updated Lagrangian formulation and automatic remeshing calculation conditions during the simulation were chosen if elements degenerate (the criterion is volume quality). The global mesh edge length was considered to be 0.8. The remeshing deformation value was also defined as 2. During the deformation, the environment temperature was set to room temperature. To define the boundary conditions for this numerical simulation step, it is assumed that the hammer (with a mass of 32 kg) moves at a maximum energy of 0.592 kJ and performs a single blow.

The constitutive equation assumed in the calculations for the tested specimens is according to the “HanselSpittel” constitutive model as shown in Equation (7):(7)$$\sigma =A\times e^{Tm_{1}}\varepsilon^{m_{2}}{\dot{\varepsilon }}^{m_{3}}e^{\frac{m_{4}}{\varepsilon }}$$
where *σ* is the flow stress, *ε* is the true strain, and *A*, *m_1–4_* are the material coefficients. The elastic parameters are: *E* = 73 GPa and *ν* = 0.3.

In addition to deformation, heat transfer and surface friction between the deformation elements and the sample during the upsetting process are also considered in the FE simulation model to more accurately reflect the experimental process. The heat transfer coefficient between the sample and environment is 10 Wm^−2^ K^−1^, and between the sample and dies is 2 × 10^4^ Wm^−2^ K^−1^.A Coulomb friction law at the tool–workpiece interface was considered the frictional boundary condition. The frictional shear stress $\tau_{fr}$ is expressed using a coefficient of friction µ as follows:(8)$$\tau_{fr}=\mu \cdot p$$

The simulations were carried out for values of the friction coefficient corresponding to the three types of friction conditions (dry, mineral oil, and graphite in oil). For the present study, the friction coefficients were 0.113, 0.055, and 0.042 obtained from experimental tests.

As shown in Figure 18 the friction phenomena, which influences plastic deformation, makes the contact surface with the upper and lower die remarkably smaller when the friction coefficient is higher. When the friction coefficient decreases the contact area increases.

To study the distribution of strains inside the sample, 3 sensors were mounted in 3 different areas: (1) at the upper tool-sample contact, (2) at the edge of the sample, and (3) on the symmetry axis of the sample according to Figure 19.

In Figure 20, equivalent strain distribution at 10% deformation is presented for studied friction conditions.

The variation of the equivalent strains according to the friction coefficient is shown in Figure 21a,b by the three sensor positions.

For all lubrication conditions, the equivalent strain has the minimum value in sensor 1 (at the tool-sample contact) where the influence of friction is maximum, with the lowest deformation value being registered in the case of the friction coefficient of 0.113.

The highest strain value was obtained in sensor 3 (the central area of the sample) where the influence of the frictional forces is minimal. Figure 22 depicts the simulated flow curves recorded in the three sensors considering dry friction deformation conditions.

Figure 23 depicts the effect of the investigated friction coefficients on the increase in temperature in the sample during the deformation process.

The temperature increase of the sample during height reduction (by 10%) is insignificant.

## 6. Discussions

The current manuscript aimed to study:
−The influence of different types of lubricants (dry, mineral oil, graphite in oil) on the friction coefficients using the ring compression test for 3 values of the strain (0.2; 0.3; 0.4);−The influence of friction conditions on the formability of aluminum alloy A 6082.

Following the experiments performed, it can be observed that:
−The degree of deformation influences the value of the friction coefficient for all 3 types of lubricants used. The friction coefficients increase significantly up to the degree of deformation of 30%, after which a very slight decrease is observed in the case of mineral oil and graphite and a significant decrease in the case of non-lubricated surfaces.−From Table 3 in the case of each type of lubricant, the deformation energy (592,000; 480,000; 320,000) determines the appearance of dimensional changes (barreling). The coefficient of barreling is the largest in the case of non-lubricated surfaces.−The use of Gubkin’s relationship to determine the coefficient of friction shows that by increasing the strain in the case of each type of lubricant, the coefficient of friction increases. In all cases, the degrees of deformation was less than 10% and the lowest values of the friction coefficients were obtained in the case of graphite lubrication, where they were between 0.023 and 0.042.−From the results presented in Table 4, the strength of the material decreases, respectively, its formability increases by reducing the coefficients of friction on the tool-sample surface. The lowest resistance value is 570.288 N/mm^2^ for graphite lubrication and the friction coefficient of 0.023.

## 7. Conclusions

By increasing the strains, the coefficient of friction also increases for all lubrication conditions.

Material deformation strength is influenced by lubrication conditions. Thus, unlubricated surfaces have higher deformation strength than mineral oil and graphite in oil-lubricated surfaces.

The non-uniformity of the strains induced by lubrication conditions was studied by deforming the sample, sectioning it, and measuring the hardness in 3 different zones (see Figure 3).

Graphite in oil was found to be more effective than the other lubricants in reducing the barreling and non-uniformity in the geometry of the samples.

Analyzing the graphs drawn for the 3 deformation areas, it was concluded that lubricants reduce friction and increase hardness. In zone 3 the material deforms the most, having the highest harnesses for lubricated surfaces and the lowest for non-lubricated surfaces. In terms of ram drop height, specimens deformed from a higher height deform more than those deformed from lower heights.

By increasing the friction coefficient, the contact surfaces sample tools significantly decrease.

By reducing the friction coefficient from 0.113 to 0.042, the strain values decrease for the same height reduction of the sample.

The numerical analysis demonstrates the fact that friction conditions significantly influence the distribution of deformations in the volume of the sample. The minimum value of the deformation is found at the contact of the sample tool because there the influence of friction is maximum, and the maximum value of the deformation is obtained in the center of the sample. The maximum strain values in the case of a friction coefficient of 0.115 are between 0.124 and 0.137 in the central area of the sample.

The values of the studied friction coefficients insignificantly influence the temperature increase inside the sample.

In future experiments, we will consider the study of the influence of the lubrication conditions on the friction coefficient for an extended number of degrees of deformation (<0.1, >0.4), so that the hypothesis of the presence of oxides acting as a lubricant on the surfaces will be verified.

## Figures and Tables

**Figure 1 materials-16-02045-f001:**
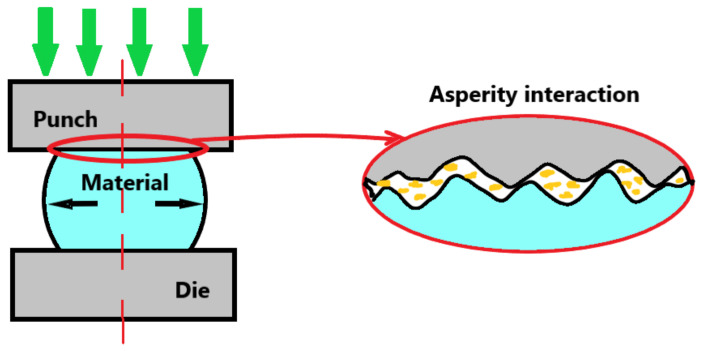
Surface roughness interaction between die and workpiece.

**Figure 2 materials-16-02045-f002:**
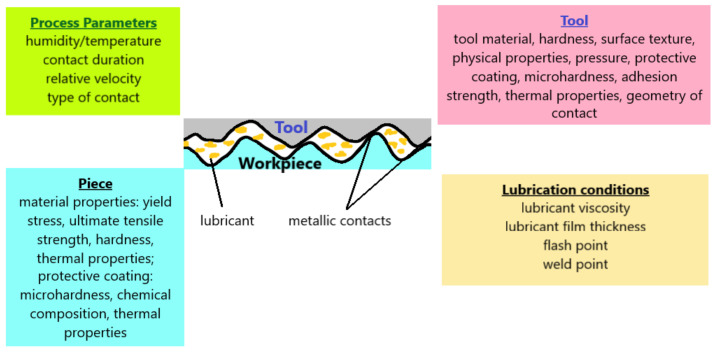
Parameters that can influence the friction in metal forming.

**Figure 3 materials-16-02045-f003:**
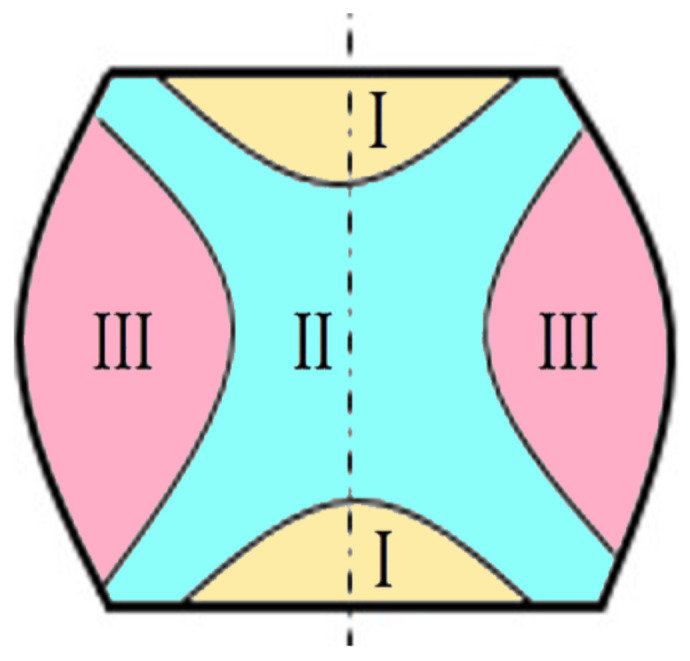
Deformation zones in upsetting (Zone I—minimum deformation; Zone II—maximum deformation; Zone III—medium deformation).

**Figure 4 materials-16-02045-f004:**
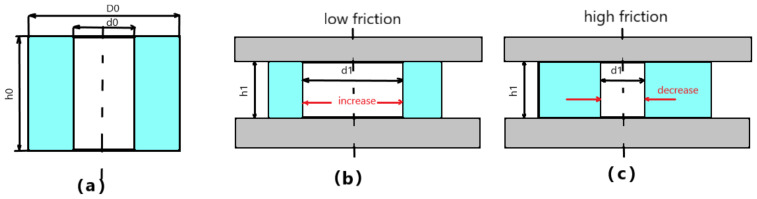
Material flow in ring compression: (**a**) initial state (**b**) low friction condition and (**c**) high friction condition.

**Figure 5 materials-16-02045-f005:**
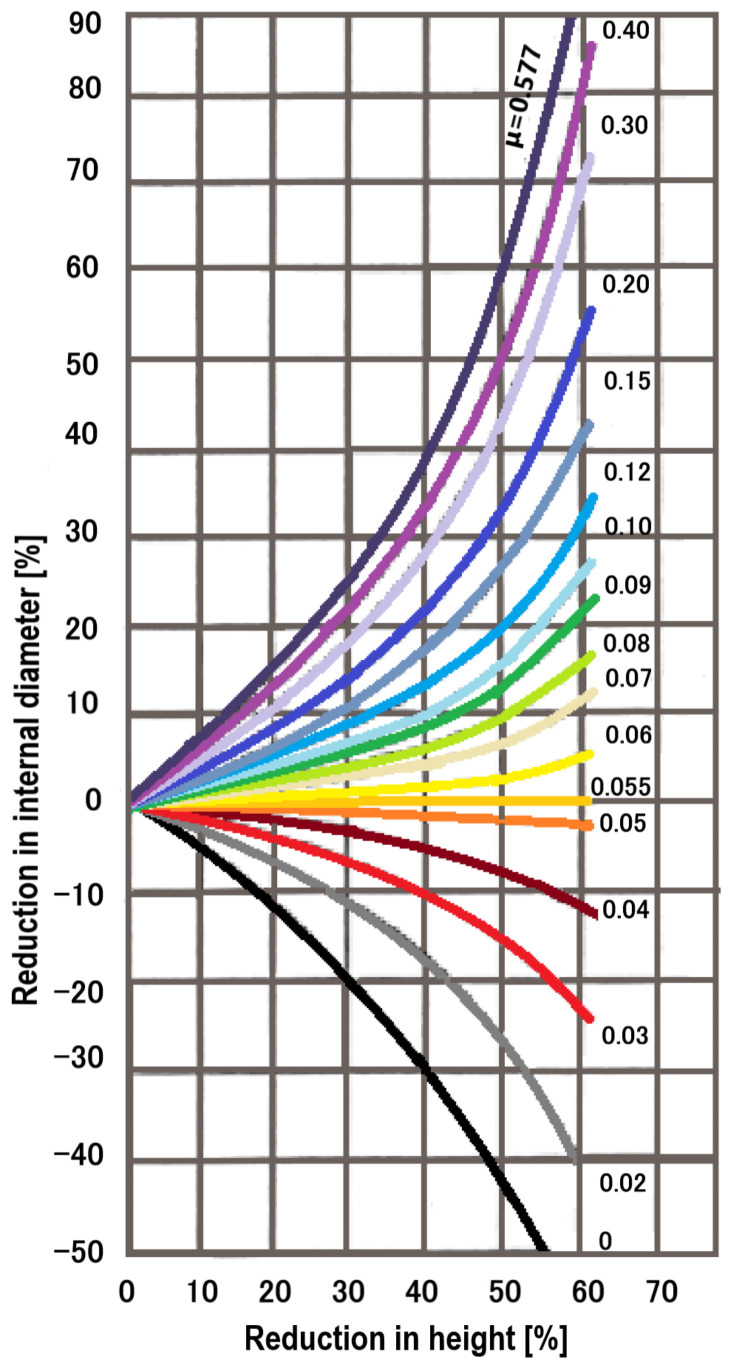
Friction calibration curves.

**Figure 6 materials-16-02045-f006:**
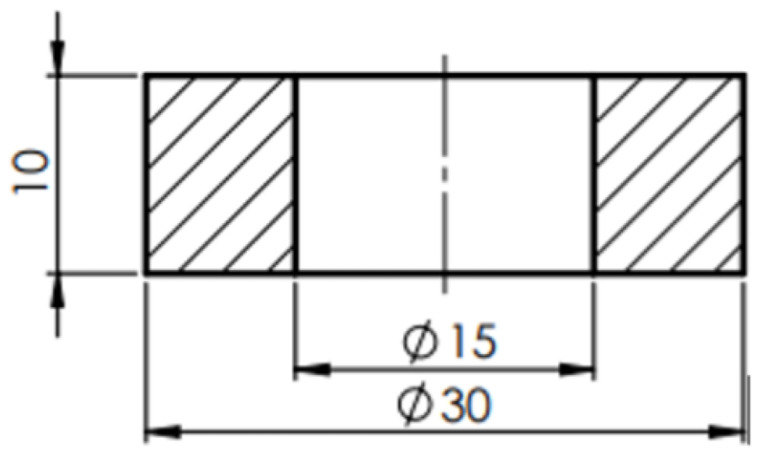
The dimensions of the ring sample.

**Figure 7 materials-16-02045-f007:**
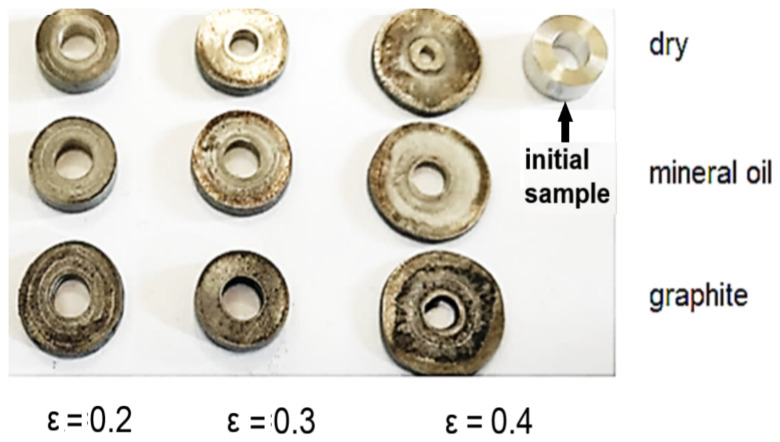
The results of applying the ring compression tests for different lubrication conditions and different strains.

**Figure 8 materials-16-02045-f008:**
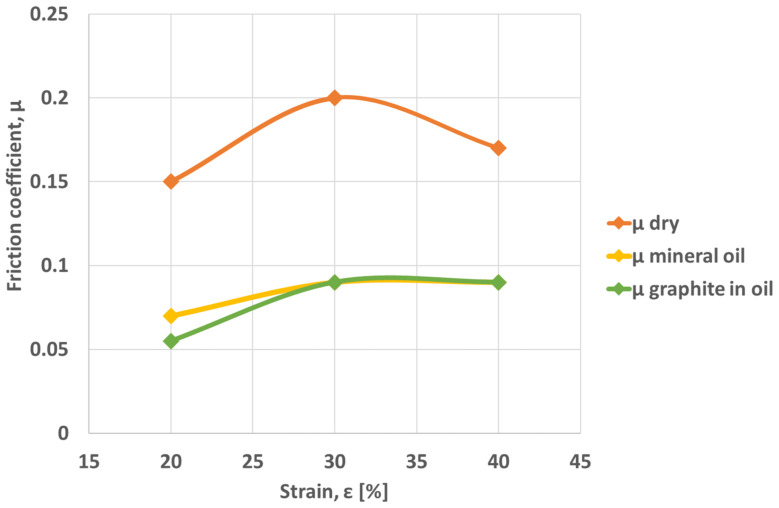
The variation of the friction coefficient vs. friction conditions and strains.

**Figure 9 materials-16-02045-f009:**
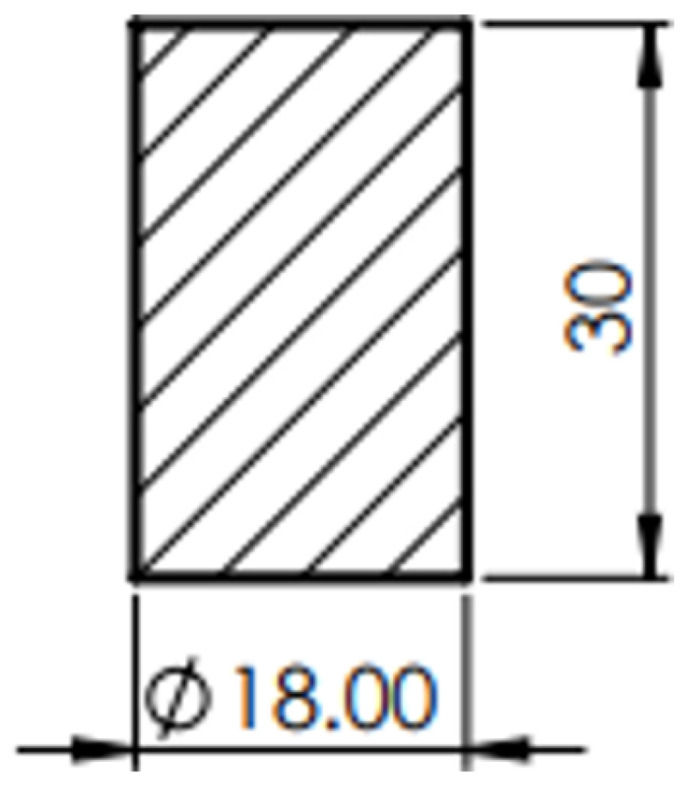
Dimensions of the cylindrical sample.

**Figure 10 materials-16-02045-f010:**
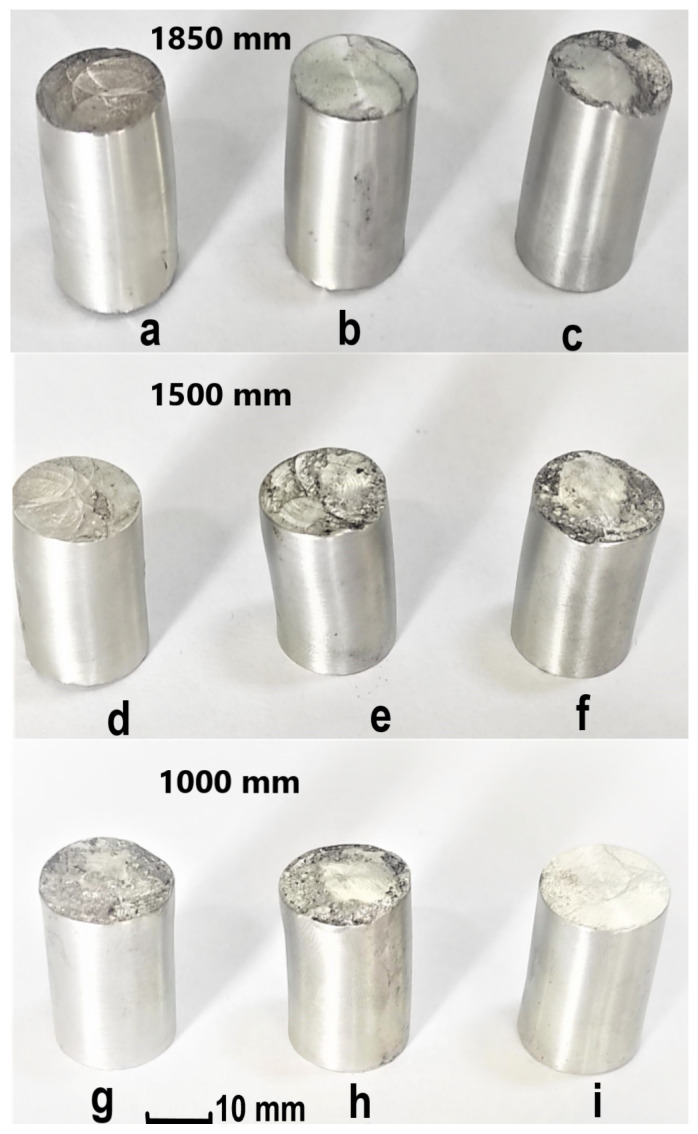
Cylindrical samples deformed on hammer from a ram drop height of 1850 mm (**a**–**c**), 1500 mm (**d**–**f**), and 1000 mm (**g**–**i**) for different friction conditions: (**a**,**d**,**g**) dry; (**b**,**e**,**h**) mineral oil; (**c**,**f**,**i**) graphite in oil.

**Figure 11 materials-16-02045-f011:**
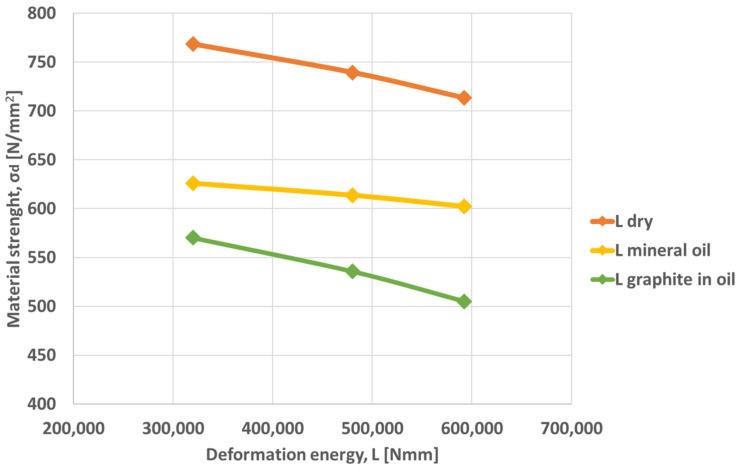
Variation of material strength vs. deformation energy for different lubrication conditions.

**Figure 12 materials-16-02045-f012:**
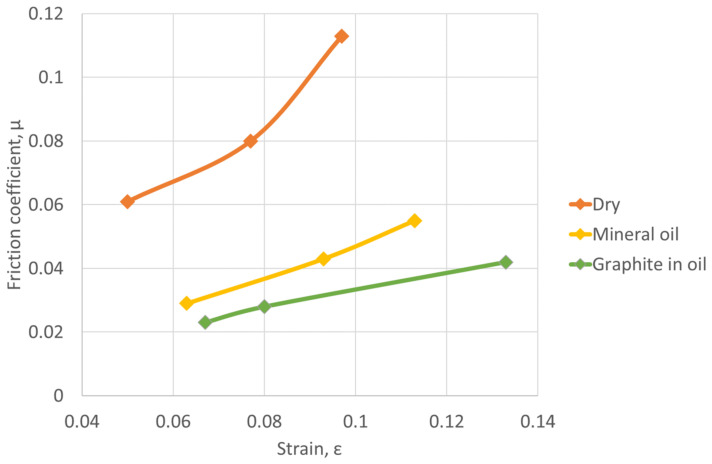
The variation of the friction coefficient vs. strains.

**Figure 13 materials-16-02045-f013:**
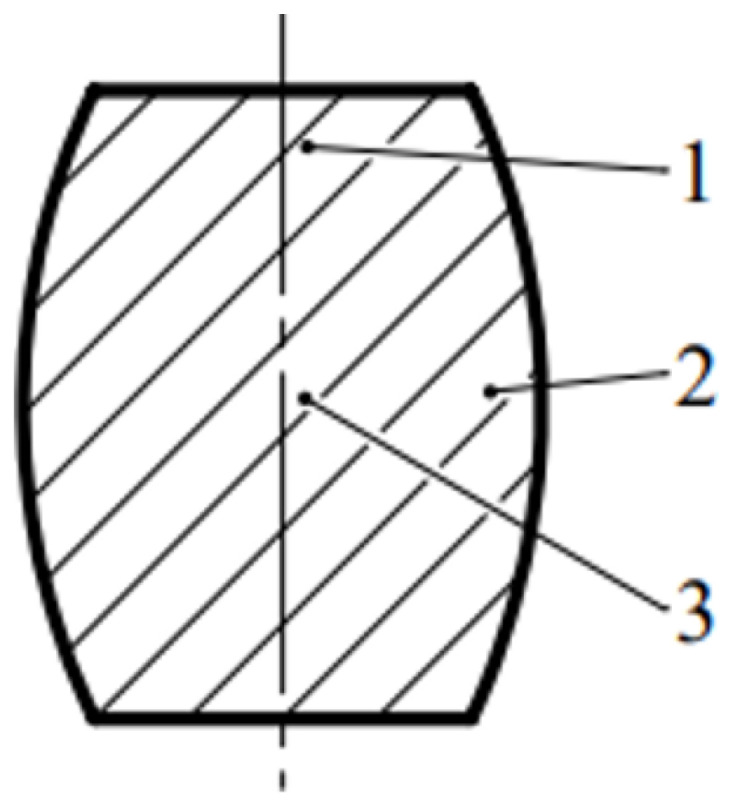
Different areas in deformed and sectioned samples: (1,2,3) points where hardness has been measured.

**Figure 14 materials-16-02045-f014:**
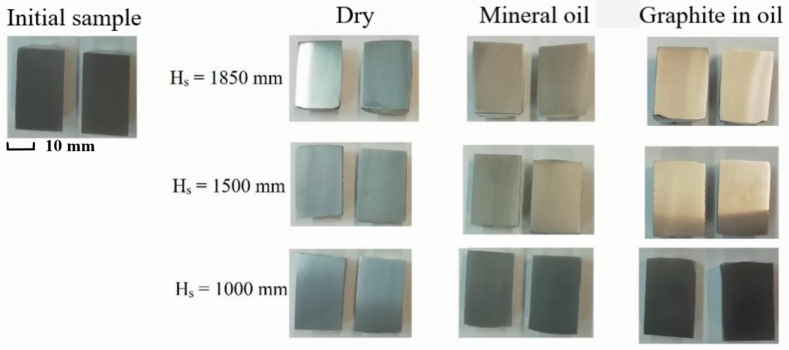
Samples were sectioned and prepared for hardness tests.

**Figure 15 materials-16-02045-f015:**
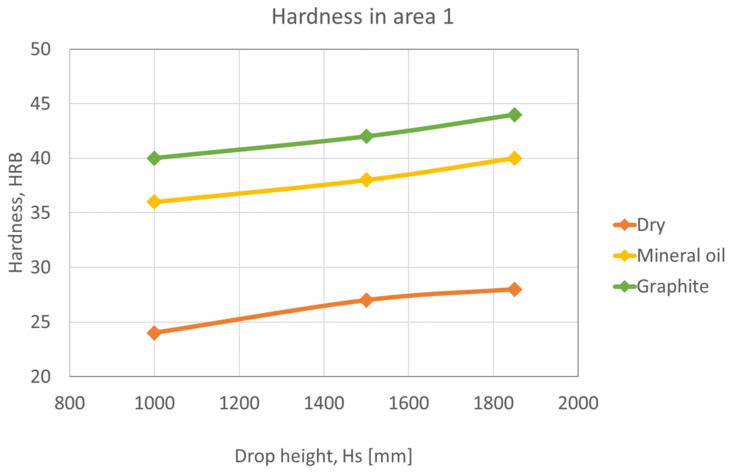
Variation of hardness as a function of ram drop height, in area 1.

**Figure 16 materials-16-02045-f016:**
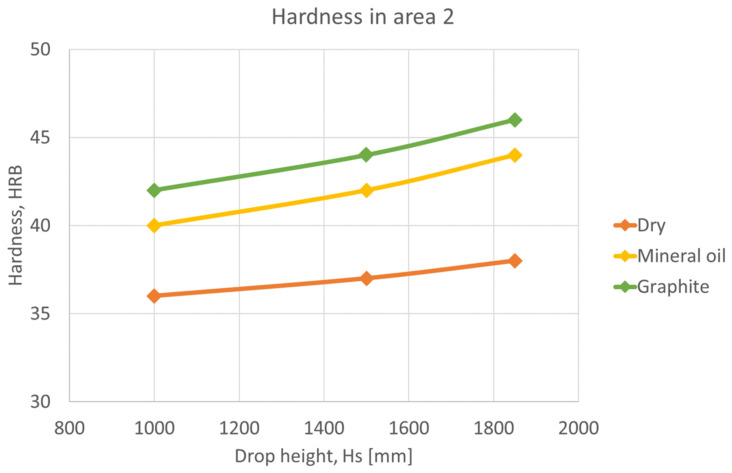
Variation of hardness as a function of ram drop height, in area 2.

**Figure 17 materials-16-02045-f017:**
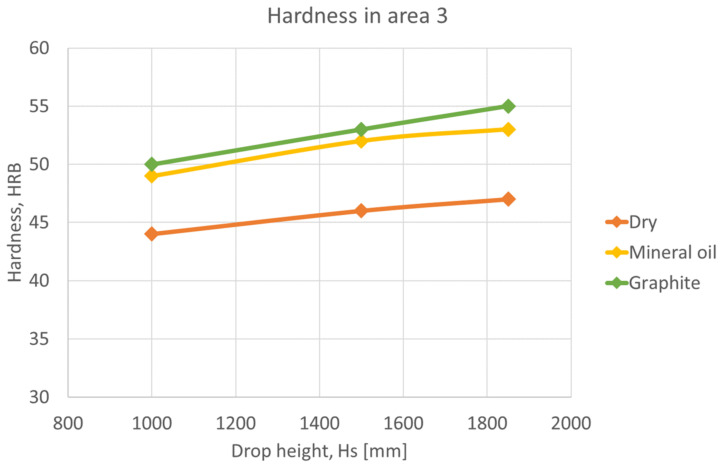
Variation of hardness as a function of ram drop height, in area 3.

**Figure 18 materials-16-02045-f018:**
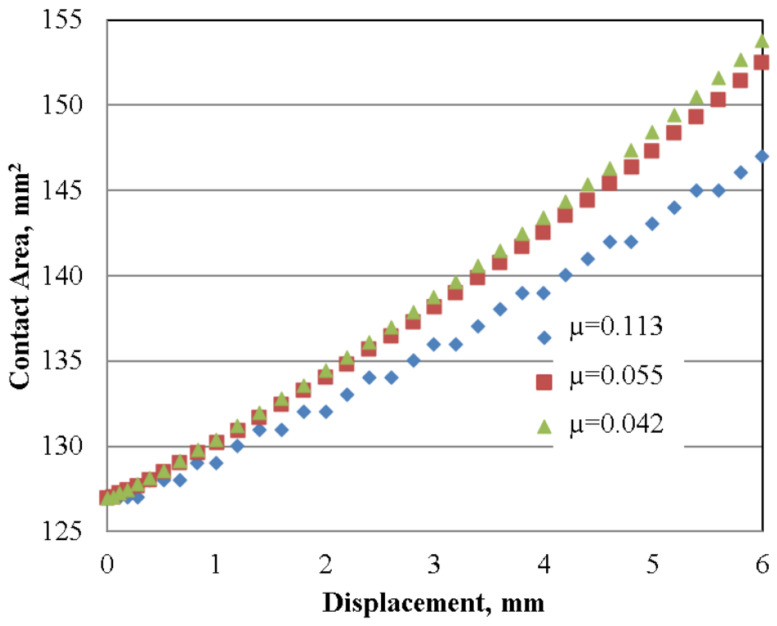
Influence of ram displacement on sample contact area for different friction conditions.

**Figure 19 materials-16-02045-f019:**
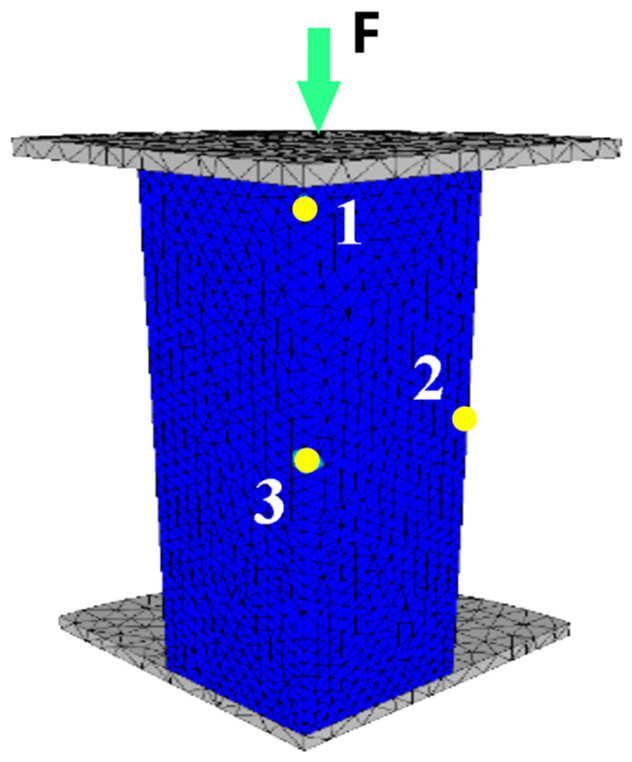
Position of the sensors in the sample: (1,2,3) the position of the sensors mounted in 3 different areas, (*F*) load force.

**Figure 20 materials-16-02045-f020:**
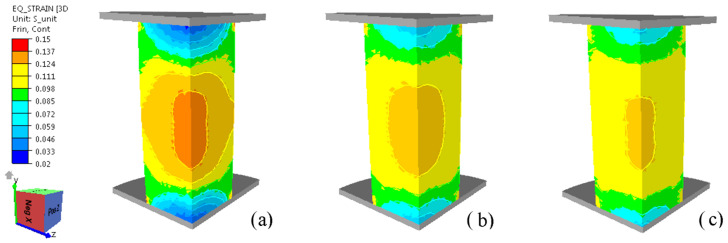
Equivalent strain distribution at 10% deformation for different friction conditions: (**a**) *µ* = 0.113; (**b**) *µ* = 0.055; (**c**) *µ* = 0.042.

**Figure 21 materials-16-02045-f021:**
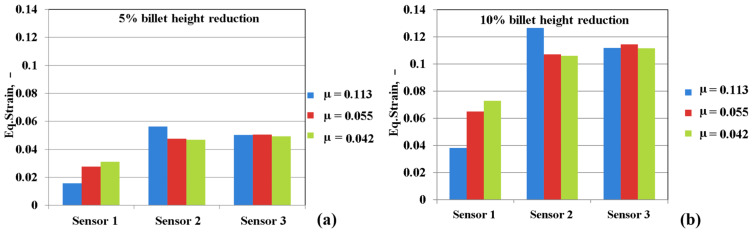
Influence of sensor positions on equivalent strain distribution for different lubrication conditions: (**a**) 5% sample height reduction, (**b**) 10% sample height reduction.

**Figure 22 materials-16-02045-f022:**
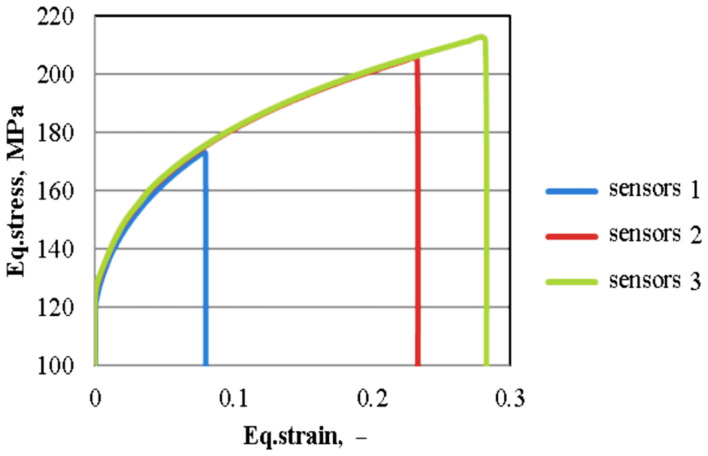
Stress-strain curves considering dry friction deformation condition.

**Figure 23 materials-16-02045-f023:**
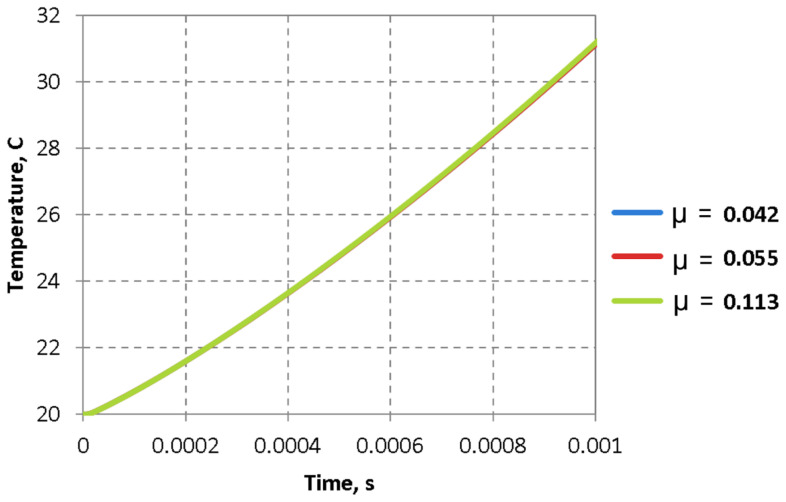
Temperature distribution in the sample during deformation.

**Table 1 materials-16-02045-t001:** % Weight composition in AA6068.

Elements	Al	Bi	Cr	Cu	Ga	Fe	Pb	Mg	Mn	Ni	Si	Ti	V	Zn
[%]	93.22–97.6	0.60–1.1	≤0.3	≤0.1	≤0.03	≤0.5	0.20–0.4	0.60–1.2	0.40–1.0	≤0.05	0.60–1.4	≤0.2	≤0.05	≤0.30

**Table 2 materials-16-02045-t002:** Experimental results of friction coefficient determinate by the Coulomb friction model.

Nr.	Friction Condition	*d_0i_* [mm]	*d_0e_* [mm]	*h_0_* [mm]	*d_1i_* [mm]	*d_1e_* [mm]	*h_1_* [mm]	*ε*	*μ*
1.	Dry	15	30	10	14.1	32.3	8	0.2	0.15
2.					13.2	34.3	7	0.3	0.20
3.					12	35.8	6	0.4	0.17
4.	Mineral oil	15	30	10	14.7	32.4	8	0.2	0.07
5.					14.2	34.9	7	0.3	0.09
6.					13.9	36.6	6	0.4	0.09
7.	Graphite in oil	15	30	10	14.9	32.18	8	0.2	0.055
8.					14.5	34.15	7	0.3	0.09
9.					14.1	36.3	6	0.4	0.09

**Table 3 materials-16-02045-t003:** Experimental results after hammer deformation.

Nr.	Friction Condition	*h_0_* [mm]	*d_0_* [mm]	*V* [mm^3^]	*H_s_* [mm]	*L* [kNmm]	*h* [mm]	*d_min_* [mm]	*d_max_* [mm]
1.	Dry	30	18	7634.07	1850	592	27.1	18.7	19.5
2.		30	18	7634.07	1500	480	27.7	18.3	19.1
3.		30	18	7634.07	1000	320	28.5	18.1	18.5
4.	Mineral oil	30	18	7634.07	1850	592	26.6	18.8	19.4
5.		30	18	7634.07	1500	480	27.2	18.6	18.9
6.		30	18	7634.07	1000	320	28.1	18.6	18.8
7.	Graphite in oil	30	18	7634.07	1850	592	26	18.9	19
8.		30	18	7634.07	1500	480	27.6	18.9	19.1
9.		30	18	7634.07	1000	320	28	18.3	18.5

**Table 4 materials-16-02045-t004:** Experimental results for different friction conditions.

Nr.	Friction Condition	*m*	*σ_d_* [N/mm^2^]	$\frac{d_{max}}{d_{min}}$	$\dot{\varepsilon }$ [s^−1^]	*δ* [−]	*ε* [−]	*µ* [−]
1.	Dry	1.069	713.544	1.043	6.412	0.041	0.097	0.113
2.		1.066	739.417	1.044	5.774	0.042	0.077	0.08
3.		1.064	768.409	1.022	4.714	0.022	0.05	0.061
4.	Mineral oil	1.071	602.132	1.032	6.412	0.031	0.113	0.055
5.		1.068	613.647	1.016	5.774	0.016	0.093	0.043
6.		1.066	625.891	1.011	4.714	0.011	0.063	0.029
7.	Graphite in oil	1.073	505.182	1.005	6.412	0.005	0.133	0.042
8.		1.068	535.747	1.011	5.774	0.01	0.08	0.028
9.		1.065	570.288	1.011	4.714	0.011	0.067	0.023

## Data Availability

Not applicable.

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
