# Peer review of "Experimental and Numerical Study on the Influence of Lubrication Conditions on AA6068 Aluminum Alloy Cold Deformation Behavior"

_materials, 2023, doi:10.3390/ma16052045_

Round 1
Reviewer 1 Report
Friction plays a significant role in the deformation process of metallic materials and determines the inhomogeneous distribution of the strain and as a result the uneven microstructure features and properties. The authors of the paper "Experimental and Numerical Study on the Influence of Lubrication Conditions on AA6068 Aluminum Alloy Cold Deformation Behavior" presents an interesting experimental investigation and finite element simulation of the deformation of aluminum alloys in the different friction conditions. The paper may be accepted for publication. However, some points of the paper should be improved accordingly following comments:
1. In the introduction part, the authors have not considered the application of the finite element method for the simulation of plastic deformation. It is recommended to analyze the papers devoted to FE modeling (e.g. works of A. Pozdniakov et al).
2. On the whole, the paper looks like a technical report without a detailed analysis of the obtained results. The discussion in the manuscript should be significantly expanded.
3. The authors wrote that “The coefficient of friction increases with the increase in the strains in all cases”. It is not correct. The friction coefficient after dry upsetting for 40 % decreases. The reason for the decrease should be given.
4. The values of friction coefficient obtained by the two methods (rings and cylinder shape samples) significantly differ. The reason for such behavior should be given. The authors applied for the calculation of the friction coefficient only one model of S. I. Gubkin. It is recommended to apply other models for the calculation of the friction coefficient using the barreling effect (10.1016/S1003-6326(15)64033-X, 10.1016/j.jmatprotec.2017.01.012, etc.)
5. The authors have used the values of the friction coefficients obtained by the upsetting of the rings for the FE modeling. It is unclear why the authors did not use the values measured by the deformation of the cylinders with more close friction conditions.
6. The calculated values of the “material strength to deformation” seem to be incorrect. The value of the stress flow should increase with a rise in the strain rate which is directly related to the potential energy of deformation. However, the authors have obtained the opposite result. It is recommended to remove the part devoted to the calculation and analysis of the “material strength to deformation” or make the comparison of the calculated stresses with the stress-strain curves obtained by FE simulation.
7. The details of the FE simulation (physical, elastic, and rheological properties of the material, mesh size, boundary conditions, etc.) should be added to the manuscript.
8. Figure 22 is not sufficient for the investigation and may be skipped.
9. Minor correction:
- As the authors write that several numbers of the sample used for the friction analysis, standard deviations should be added to the values of the friction coefficients and hardness in Tables and Figures.
Reviewer 2 Report
The manuscript has the subject "Experimental and Numerical Study on the Influence of Lubrication Conditions on AA6068 Aluminum Alloy Cold Deformation Behavior" and can be published after some revisions:
1. During cold deformation the temperature of the specimens increases. How does this heat affect the results?
2. What influence do lubricants have on the presence of micro-defets of the specimens?
3. Some impurities from the lubricants can get into the specimens during deformation. How does this affect the final properties?
4. The following manuscript shows that cold deformation increases the dislocation density near grain boundaries. What influence do lubricants have on the dislocation density?
https://doi.org/10.1016/j.ijhydene.2021.09.013
Reviewer 3 Report
The paper must necessarily include nomenclature - all symbols, abbreviations and markings must be presented there - they are not fully explained in the text - already at the beginning of paragraph 2, not everything is explained and described. These are things that need to be made clear, especially in journals of renown Materials MDPI.
Authors must carefully follow the text of the paper.
Abstract must provide basic information about what is in the manuscript - briefly succinctly, without referring to any results - it must be shortened and corrected, without giving any symbols, abbreviations and literature references.
The word "sample" should not be used for the material used in testing and research - the word "specimen" is preferred, as stated in ASTM or BS standards.
The literature review is correct.
In relation to scientific papers, the word "work" should not be used - the words "paper, scientific paper, manuscript, scientific article" are preferred.
I recommend making the figures yourself, and not copying them from the literature - these are not difficult graphics to do in vector graphics programs. We are talking about figures 1,2,3,4.
In table 3, in the column where the value L [Nmm] is given, I recommend changing the order of magnitude - give it in other units, maybe [J] or [kNmm] - this will allow you to leave zeros in the notation - it will be clearer in notation and understanding for the reader.
I recommend changing the colors in tables from 3 and on - the paper will be more readable.
I suggest that from graphs 11 to 17, where the authors show changes in subsequent quantities as a function of various parameters, draw trend lines, along with appropriate mathematical formulas, and provide the coefficient of determination, which will allow drawing appropriate conclusions.
In the section devoted to numerical calculations, please complete the text of the thesis with information on the number of nodes in the model, specify what finite elements were used in the simulation, how many nodes, numerical integration points and Jacobian points were in one finite element. Whether the finite elements used were three-dimensional or flat. Please show the numerical model with the assumed boundary conditions and indicate where the load was applied. Please complete the manuscript by providing information on the convergence of the numerical model - what was the size of the finite element, what were the subsequent dimensions of the finite elements used. What dictated the selection of the finite elements used and the division of the model into these elements. Please refer in your paper to the issue of convergence of numerical estimation - what was the selection criterion? Please indicate what material model the authors assumed in the calculations for the tested specimens - what were the properties of the model - please also provide the tensile curve.
Please indicate what dictated the selection of subsequent sensors. Please complete the manuscript with a figure indicating some reference dimensions for these sensors from specific planes or points. It is necessary.
Some of the figures “are hard to read” - it is worth correcting them. For graphs, I recommend changing the application from MS Excel to GRAPHER, ORIGIN or another.
Please correct the manuscript, complete it and submit it for re-review. I suggest a major revision.
Round 2
Reviewer 1 Report
The authors have answered previous comments and improved the manuscript. The paper may be accepted for publication.
Reviewer 3 Report
The authors included almost all my suggestions in the revised version of the paper. I recommend the manuscript for publication.